# What's Black and White and Pink All Over? Lesser Flamingo Nocturnal Behaviour Captured by Remote Cameras

Paul E. Rose [1,2,]*[ID], Jess Chapman [3], James E. Brereton [3][ID] and Lisa M. Riley [4][ID]



1   Centre for Research in Animal Behaviour, Department of Psychology, University of Exeter, Exeter EX4 4QG, UK
2   WWT, Slimbridge Wetland Centre, Slimbridge, Gloucestershire GL2 7BT, UK
3   University Centre Sparsholt, Sparsholt, Winchester SO21 2NF, UK
4   Centre for Animal Welfare, University of Winchester, Winchester SO22 4NR, UK
*   Correspondence: p.rose@exeter.ac.uk

**Abstract:** The study of animal behaviour is important for the development of husbandry and management practices for zoo-housed species. Yet, data are typically only collected during daylight hours, aligning with human work schedules rather than animal activity patterns. To remedy this, 24 h data collection is needed. This study investigated the behaviour of a captive flock of lesser flamingos to understand temporal changes in their time-activity patterns. Two remote camera traps were placed around the birds' outdoor enclosure and one within the indoor house. Counts of birds visible within specific enclosure zones were recorded from photographic data. Behaviour was defined as active or inactive, and modified Spread of Participation Index (SPI) was used to calculate enclosure zone occupancy. Results indicated that lesser flamingos are active overnight, and to a similar amount as in the daytime. Proportions of birds observed as active were significantly higher at later times of the day (i.e., dusk) when compared to the number of active birds in the morning. Enclosure usage was diverse and indoor and outdoor zones could be used by different numbers of birds at different times of the day. Variation in enclosure usage may indicate the changing needs of the flamingos when housed indoors overnight and when they have night-time access to an outdoor enclosure. This research has identified the need for further research into the nocturnal behaviour and space use of lesser flamingos and suggests the need for 24 h research in captive birds, and other zoo-held species, especially when species are locked indoors or face behavioural restriction overnight due to biosecurity measures surrounding zoonoses outbreaks, e.g., *Avian Influenza*.

**Keywords:** lesser flamingo; welfare; nocturnal behaviour; remote monitoring; husbandry evidence





## 1. Introduction

Behavioural study may be the commonest form of research undertaken in zoos and aquaria [1] and behavioural data are used to answer a wide range of zoo-centric research questions [2]. Whilst daytime study of time-activity budgets is common, night-time behaviour patterns are less often researched [3]. Although often considered diurnal, many zoo-housed species can have specific crepuscular and/or nocturnal activity patterns. One such group of animals are the flamingos (*Phoenicopteriformes*). Previous research conducted in the wild and in captivity has identified a diverse range of behaviours performed during early morning, late evening and overnight [4–8], for example differences in foraging activity and style of foraging, and movement patterns. In other taxa, data collection across a complete 24 h cycle has yielded relevant evidence for the reassessment of husbandry and management to promote behavioural diversity, reduce abnormal behaviour patterns, and improve welfare [9–13]. For example, when given access to their outdoor enclosure overnight, captive female Asian elephants (*Elephas maximus*) showed increased play and decreased swaying behaviour, suggesting a change in management strategy is needed to improve welfare in these animals [14]. As assessment of zoo bird behaviour across a

24 h period is particularly poorly studied, this paper aimed to build on the suggestion of Rose et al. [15] that study of captive flamingo activity after dark would be useful to greater understanding of the needs of zoo-housed flamingos; especially given that wild flamingos remain active overnight to perform feeding, vigilance and movement behaviours.

Patterns of foraging behaviour in greater flamingos (*Phoenicopterus roseus*) closely aligned with blooms of *Artemia spp.* (brine shrimp) across a 24 h period [5], with flamingos only able to forage nocturnally due to the vertical migration of *Artemia* up to the water's surface (and within reach of the flamingos) during hours of darkness. Caribbean flamingos *(P. ruber)* also displayed nocturnal foraging patterns and the amount of time spent feeding changed with prey availability and flock size [4]. However, whilst nocturnal foraging is clearly an important mechanism used by flamingos to meet daily energy needs, this behaviour may be less efficient than diurnal foraging. Beauchamp and McNeil [16] showed that nocturnally foraging Caribbean flamingos performed more vigilance scanning (for predators) at the expense of feeding time. Physiological change in flamingos may also be a driver for nocturnal feeding. Further research on the ecology of the Caribbean flamingo, specifically the population of this species that occurs in the Galápagos Islands (Ecuador), suggested that higher rates of nocturnal foraging may correspond with the provisioning of a chick with parent-manufactured crop secretion ("crop milk") [7], but other factors, such as the quality and profitability of foraging patches may also influence nocturnal feeding. As production of crop milk requires direct investment of energy from the parent flamingo [17], breeding birds may resort to more nocturnal foraging to maintain their own energetic demand plus that required to secrete regular supplies of crop milk for their chick. Tindle et al. [7] noted parent Caribbean flamingos engage in a range of nocturnal activities, suggesting a physiological driver of nocturnal foraging in breeding flamingos.

Captive greater flamingos showed increased rates of active behaviours in the later afternoon (from 15:00), early evening (from 18:00 to 20:00) and early morning (around 04:00 and 05:00) [6]. This research also identified significant differences in enclosure zone usage, whereby birds used pool areas more overnight and terrestrial areas more during daylight hours. Such findings are important for the development of best practice husbandry guidelines as well as for optimal indoor housing as access to water overnight is clearly important to natural behaviour performance (and therefore bird welfare).

To build upon the findings of Rose et al. [6] and extend our knowledge of nocturnal behaviour across the different species of flamingo, we studied the highly specialised lesser flamingo *(Phoeniconaias minor)*; a specialist in terms of habitat selection and foraging niche, bill anatomy, social behaviour and reproductive strategy [18,19]. The lesser flamingo is the smallest species of flamingo [17] and the most gregarious, with aggregations of birds noted into the millions in the East African Rift Valley [20]. This species is listed as Near Threatened [21] due to climatic change and anthropogenic factors detrimentally affecting the small number of regular feeding and breeding grounds used by this species in the wild [22]. The lesser flamingo is not commonly housed in zoos, but their welfare is as important as any other species.

This study aimed to evaluate significant differences in lesser flamingo activity at different time periods of a full 24 h period. Thus, behavioural data on the time-activity of long duration state behaviours [23] was analysed, along with space utilisation of the lesser flamingo flock at the Wildfowl & Wetlands Trust (WWT) Slimbridge Wetland Centre UK.

We consistently recorded flock behaviour to compute a 24 h time-activity budget, and based on wild literature, we hypothesised that the number of birds recorded as foraging and moving would not be significantly different between diurnal and nocturnal time periods. Thus, we predicted that flamingo activity would remain high during periods of darkness, as has been noted in wild birds and in other captive flamingo species. We further predicted that enclosure usage would not vary significantly between morning, day, afternoon, evening and overnight.

## 2. Materials and Methods

Ethical approval for the project was provided by the Ethics Committee of University Centre Sparsholt and after review by WWT living collections teams and veterinary department as part of WWT's Animal Welfare & Ethics Committee.

### 2.1. Sample Population, Study Location and Camera Placement

Data were collected on the lesser flamingo flock (n = 43) at WWT Slimbridge from January to July 2018. All birds were adult, 25 were male and 18 were female, and no incidence of ill health or injury were evident during the data collection period. All birds were flight restrained by pinioning or feather trimming due to the outdoor enclosure being open topped. No nesting occurred during the data collection period although birds did perform courtship display. Three Denver 1080 p 8 MP night-vision motion-activated cameras were positioned in the lesser flamingo enclosure. These cameras had 48 infrared LEDs for crisp night vision image capture and a passive infrared sensor (PIR) movement sensor to automatically capture a moving object within a 25 m range. For data collection, image file capture was set to the highest setting (8-megapixel photographs).

One camera was located within the flamingo's indoor house while two cameras were used to photograph the outdoor enclosure; these were fixed onto perimeter fencing. Figure 1 provides an enclosure map that shows camera locations. The outdoor camera on the far left was approximately 21 m distance from the centre enclosure island and the outdoor camera on the far right was approximately 16 m distance from the centre island. The indoor camera was fixed on a wall and as the Lesser Flamingo House was approximately 18 m in length, birds were always in range to motion activate the camera.

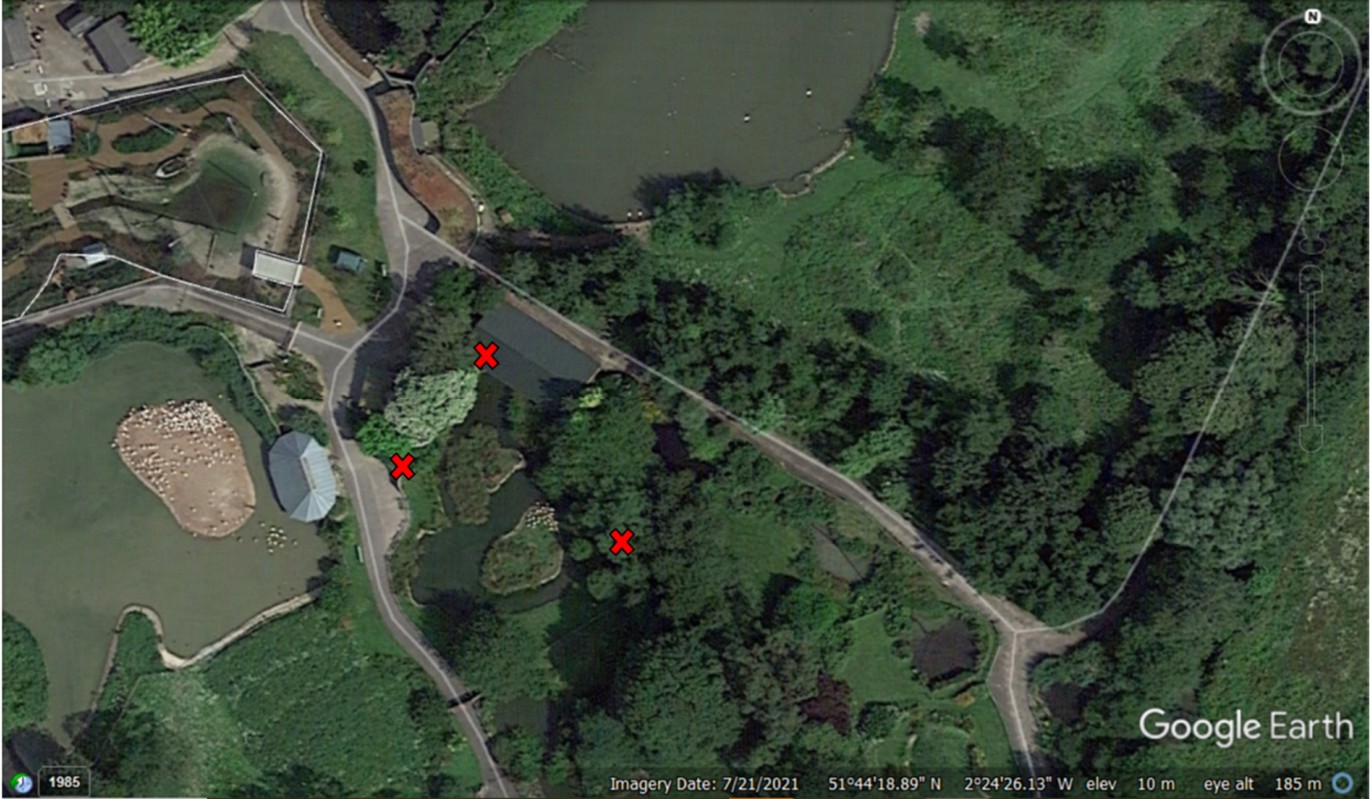

**Figure 1.** Map of the lesser flamingo enclosure at WWT Slimbridge with red crosses showing the placement of the three remote cameras (one inside the house and two covering the outside enclosure). Flamingos can be seen as white dots on the centre island (the landscape feature used for judging the range of motion activation for each outside camera).

Cameras recorded a photograph every five minutes. Outdoor and indoor behaviour and enclosure usage was examined separately. The average proportions of active and inactive birds were compared for the outdoor and indoor enclosure. At the time, the Lesser Flamingo Pen at WWT Slimbridge covered approximately 1053 m$^2$ with the indoor house being approximately 209 m$^2$. The outdoor enclosure was approximately 32 m at its widest point and 40 m at its longest. The outdoor enclosure consisted of a large pool surrounded by separated areas of grassed land, two islands, and a patch of rushes to encourage natural foraging behaviour. The flamingos also had access to a large house which contained an indoor pool and was maintained at 15–20 °C. Birds were fed twice a day, at around 08:30 and 15:00, with both feeds occurring indoors in the feeding pool. Public viewing was at the front of the outside enclosure, looking over the front grassed area over to the bird's pool. Indoor public viewing was through a set of windows down one side of the bird's house.

## 2.2. Enclosure Usage and Behavioural Data Collection

To record enclosure usage, the outdoor enclosure and indoor house were split into zones based on resources accessible to the flamingos. The modified Spread of Participation Index (SPI) was then used for calculation of overall enclosure usage [24] to take into account the zones of unequal areas. SPI provides a comparison of observed against expected frequency of occupation for each zone and provides a minimum value of 0 (equal occupancy of all zones) and maximum of 1 (unequal occupancy of zones) [24]. The indoor house was split into three zones- feeding pool (5% of total space), indoor pool (40%) and indoor dry land (55%). The outdoor enclosure was split into seven zones consisting of a smaller sanded island (3% of total space), larger nesting island (10%), rushes (5%), pool (36%), grassed land at the rear of the enclosure (10%), grassed land at the front of the enclosure (17%) and grassed land to the right of the main pool (19%). Occupancy of grassed areas was analysed separately (rather than having grassed land as one overall enclosure zone) due to different factors influencing their use. The back grassed area was narrow and close to mature trees and therefore it was less exposed to direct sunlight. The front grassed area was closest to visiting public. The grassed area to the right of the pool was largest, widest, and flattest, allowing all the flock to congregate and interact together.

Behavioural data gathered from each photograph were categorised for analysis as Active (foraging, moving, courtship and preening) and inactive (resting, sleeping, standing). For a subsample (1502 records, 29% of recordings) of outdoor photos where bird behaviour could be reliably identified more specifically than simply active or inactive, analysis of specific behaviours (foraging, moving, preening, resting and courtship) within each zone and at different times of day was undertaken.

Foraging was defined as a bird with head in the water, filtering for food, and either wading or swimming whilst filtering. Foraging included consumption of flamingo pellet and natural food. Moving was defined as the bird walking or running on two legs, or swimming in the water without filter feeding. Courtship was defined as the ritualised group display that is species-typical for lesser flamingos, including marching and wing saluting (see Kahl [25] for more detail). Preening was defined as the cleaning, oiling, and re-arranging of feathers using the bill. Inactive behaviours (resting, sleeping, and standing) were defined as a bird displaying limited to no movement, potentially with head "tucked under wing" or standing with no other apparent body movement.

To calculate the proportion of birds active, the number of birds performing an active behaviour was divided by all birds visible. Birds labelled as displaying an "Unknown" behaviour were not included in any behavioural analyses but were included in the enclosure usage analysis (as it could be discerned that a flamingo was in a specific enclosure zone even if its actual behaviour was not identifiable). Where uncertainty was present, nearby birds were used to aid identification- for example, one bird with a wing outstretched was likely preening, while six birds with wings outstretched suggested a courtship behaviour.

Observations were categorised into specific times of day during data collection. The codes used were 'Early', 'Dawn', 'Morning', 'Afternoon', 'Dusk' and 'Night'. These were

subsequently recategorized for analysis based on light levels: 'Day' for the daylight hours, 'Twilight' for the hours of dawn and dusk, and 'Dark' for the hours of darkness. The cut-off times for each time code were set per month and defined as per Table 1. Keeping time codes dynamic helped to control for seasonality and changes in day length with season. The descriptions in Table 1 were taken from https://www.timeanddate.com (accessed on 1 September 2018).

**Table 1.** Definitions for each time code when taking behavioural data from the camera trap photos.

| Category | Time Code | Definition |
| --- | --- | --- |
| Dark | Early | Between 00:00 and one minute before the earliest morning civil twilight recorded for that month. |
| | Night | From one minute after the latest civil evening twilight recorded that month until 23:59. |
| Twilight | Dawn | Between the earliest morning civil twilight and the latest sunrise recorded for that month. |
| | Dusk | Between the earliest sunset and latest evening civil twilight recorded for that month. |
| Day | Morning | Between one minute after the latest sunrise and mid-month solar noon (usually taken as the 15th of the month except for February (14th)). |
| | Afternoon | Between 'Morning' and 'Dusk'. |

### 2.3. Housing and Avian Influenza Precautions

Cameras in the indoor house and outdoor enclosure could not be set up for recording concurrently. Due to an *Avian Influenza* outbreak in the UK over the winter of 2017/2018, the flamingos were housed indoors full-time from January until late March. Indoor photos were taken during this period as the camera could be accessed without disruption to the flamingos. Biosecurity measures were followed when entering the flamingo house to access the camera. Once AI restrictions were lifted, birds had free choice access to both the indoor house and outdoor enclosure. However, due to this period of indoor-only housing, photographs from the inside camera and from outside cameras were analysed separately for the whole project. Bird behaviour was sampled indoors from 30 January 2018 to 23 July 2018, and outdoors from 29 March 2018 to 20 July 2018.

### 2.4. Data Analysis

Data were analysed using R v. 4.1.0 [26] on the RStudio v. 2022.07.1 platform [27] (specifically for application of mixed effects models to deal with repeated measures) and in Minitab v. 21.2 [28] for Poisson regression, one-sample sign tests and the creation of boxplots. For assessment of model fit, the package "MuMin" [29] was used to generate $r^2$ values for each model. Any applicable post hoc testing for significant predictors was undertaken using the "lsmeans" [30] and "pbkrtest" [31] packages using a pairwise comparison in RStudio. Plots of residuals against fitted values were examined using the "plot (model name)" code for overdispersion and were judged suitable for testing to continue. Alongside of this, the package "performance" and code "model.check (model name)" was used to review the quality of the model fit and the overall relevance of predictors [32]. To remove any likely collinearity of variables, Variance Inflation Factors (VIFs) were calculated and reviewed. VIFs above 2 were considered to show multicollinearity and would be removed from the model. VIFs were calculated using the "car" package in RStudio for linear mixed effects models; for predictors in the outdoor activity model, VIFs were 1.02; VIFs of 1.14 for indoor activity; for outside SPI VIFS were 1.03, and for indoor SPI, VIFs were 1.11. VIFs were automatically calculated by Minitab for the Poisson regression and the range of VIFs was 1.0 to 1.42 across all Poisson Regression models run.

The raw dataset for this project is available to download at https://doi.org/10.6084/m9.figshare.21546825 (accessed on 2 November 2022).

To determine any difference in flamingo activity observed across all observations (758 outdoor observations and 927 indoor observations), a linear mixed effects model was run using the "lmerTest" package in RStudio [33], with observation date as the random factor. The time of day (coded as per Table 1) and the month of data collection were included as predictors and the proportion of active birds in each photo was the outcome variable. Satterthwaite's method was used to generate the F value for each predictor using the "anova (model name)" code.

For the subset of observations where specific behaviours were accurately identifiable, any significant differences in the number of birds seen performing behaviour in the outside zones at different times of the day was analysed using a Poisson regression. The total count of birds seen in each enclosure zone for each time code for each month of the study were used for this regression. Season (as a predictor) was also first included but this showed high collinearity with month and was removed. The interaction between time code and enclosure zone was also included in the model. The zone "Water", the time code "Afternoon" and the month "May" were used as the reference levels for the model. The final regression model run was time code + month + enclosure zone + enclosure zone*time code. Analysis of variance for each predictor for each behaviour were determined via Wald's Chi-squared tests.

RStudio was again used for the same repeated measures linear mixed effects modelling (lmerTest) to determine any relationship between month and time of day categories (as predictors) on indoor and on outdoor SPI values (as outcome variables). Date was included as a random factor

Opportunistically, there were limited recordings of when a keeper was present within the indoor enclosure. The husbandry regime of the flamingos required a keeper to enter their indoor housing and these data were analysed separately to understand any potential influence of keeper presence on flamingo enclosure usage and activity, particularly as the birds were locked inside the indoor house due to the *Avian Influenza* outbreak. To describe any potential impact of the keeper's presence on flamingo behaviour and house zone usage, one-sample sign tests were run to compare the median SPI or median proportion of birds active with and without a keeper. To deal with the limited number of observations, the median proportion of active birds or median SPI with a keeper present was compared to the normal situation of no keeper being present (the majority of datapoints).

## 3. Results

### 3.1. Number of Flamingos Seen Active Compared to Inactive

Results from the camera trap images showed that these lesser flamingos were active throughout both the night and daylight hours. When birds had access to the outdoor enclosure, more birds were active at night and in the early hours (post-midnight) and were most likely to be inactive at dawn. When flamingos were recorded indoors, activity was again highest overnight and inactivity towards and during dawn (Figure 2).

Regarding outdoor recordings, there was a significant effect of time of day on the proportion of flamingo's observed as active ($F_{5, 743.5}$ = 13.83; $r^2$ = 22.5%; $p$ < 0.001) but there was no effect of month ($F_{4, 59}$ = 0.831; $r^2$ = 22.5%; $p$ = 0.511). The comparison of the proportion of birds seen active during different time categories are provided in Table 2. *P*-values are compared to a corrected alpha level [34] of 0.03 and adjusted Q values of significance are provided. Flamingos were more active at the Early (post-midnight) period and Dusk compared to Morning when outside. Nocturnal patterns of activity were also seen when birds were indoors. However, flamingo activity seemed to be higher for Afternoon hours when birds were inside, and this may be influenced by husbandry routines.

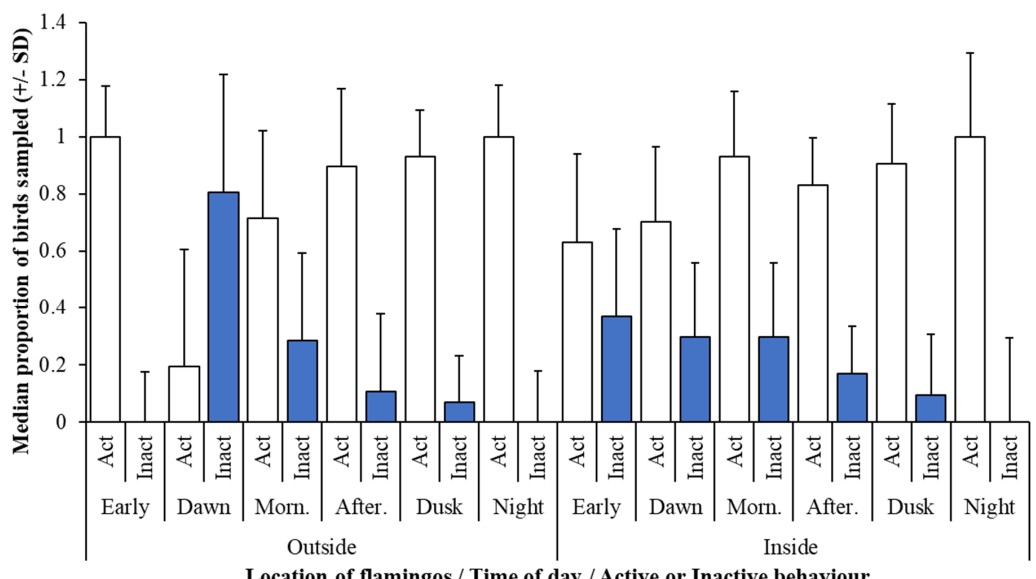

**Figure 2.** Median proportion of birds observed as active or inactive for the total number of observations from six different times of the day both in the outdoor enclosure and in the indoor house. Morn. = Morning time code and After. = Afternoon time code. Abridged on the figure for clarity of the X axis.

**Table 2.** Post-hoc comparison of outdoor activity by time of day. Significant differences are marked with an asterisk.

| Comparison | Estimate | SE | df | t Ratio | *p*-Value | Q Value |
|---|---|---|---|---|---|---|
| Afternoon–Dawn | 0.400 | 0.117 | 747 | 3.425 | 0.009 | 0.030 * |
| Afternoon–Dusk | −0.089 | 0.003 | 734 | −3.446 | 0.008 | 0.027 * |
| Afternoon–Early | −0.119 | 0.055 | 740 | −2.136 | 0.269 | 0.040 |
| Afternoon–Morning | 0.124 | 0.027 | 684 | 4.629 | <0.001 | 0.003 * |
| Afternoon–Night | −0.113 | 0.038 | 748 | −2.970 | 0.036 | 0.033 |
| Dawn–Dusk | −0.489 | 0.118 | 748 | −4.141 | <0.001 | 0.007 * |
| Dawn–Early | −0.519 | 0.127 | 748 | −4.071 | <0.001 | 0.010 * |
| Dawn–Morning | −0.276 | 0.119 | 746 | −2.326 | 0.185 | 0.037 |
| Dawn–Night | −0.513 | 0.121 | 748 | −4.247 | <0.001 | 0.013 * |
| Dawn–Early | −0.029 | 0.059 | 736 | −0.498 | 0.996 | 0.047 |
| Dusk–Morning | 0.213 | 0.033 | 722 | 6.445 | <0.001 | 0.017 * |
| Dusk–Night | −0.023 | 0.041 | 740 | −0.564 | 0.993 | 0.043 |
| Early–Morning | 0.242 | 0.059 | 745 | 4.134 | <0.001 | 0.020 * |
| Early–Night | 0.006 | 0.065 | 744 | 0.089 | 1.00 | 0.050 |
| Morning–Night | −0.237 | 0.043 | 730 | −5.474 | <0.001 | 0.023 * |

For indoor activity, both month ($F_{6, 65.7} = 4.02$; $r^2 = 15.5\%$; $p = 0.002$) and time of day ($F_{5, 764.6} = 6.279$; $r^2 = 15.5\%$; $p < 0.001$) had showed a significant relationship with bird activity. Post—hoc comparison of month and time effects on the proportion of birds seen active are provided in Table 3. Corrected alpha levels for time of day (0.013) and for month (0.005) were applied and significant Q values highlighted. Flamingos were more likely to be inactive indoors during the afternoon when compared to the break of day and during hours of darkness (Early). Overall, flamingos were more active at Dusk compared to Dawn.

**Table 3.** Post-hoc comparison of indoor activity by time of day and by month of observation. Significant differences are marked with an asterisk.

| Comparison | Estimate | SE | Df | t Ratio | *p*-Value | Q Value |
|---|---|---|---|---|---|---|
| Time | | | | | | |
| Afternoon–Dawn | 0.159 | 0.035 | 761 | 4.496 | <0.001 | 0.003 * |
| Afternoon–Dusk | 0.012 | 0.029 | 765 | 0.408 | 0.999 | 00.037 |
| Afternoon–Early | 0.119 | 0.034 | 766 | 3.534 | 0.006 | 0.010 * |
| Afternoon–Morning | 0.012 | 0.018 | 762 | 0.663 | 0.986 | 0.033 |
| Afternoon–Night | 0.009 | 0.038 | 757 | 0.256 | 0.999 | 0.040 |
| Dawn–Dusk | −0.147 | 0.043 | 764 | −3.464 | 0.007 | 0.013 * |
| Dawn–Early | −0.040 | 0.047 | 764 | −0.865 | 0.955 | 0.030 |
| Dawn–Morning | −0.147 | 0.036 | 758 | −4.057 | <0.001 | 0.007 * |
| Dawn–Night | −0.149 | 0.049 | 766 | −3.006 | 0.033 | 0.020 |
| Dusk–Early | 0.107 | 0.041 | 766 | 2.589 | 0.101 | 0.023 |
| Dusk–Morning | 0.0003 | 0.029 | 766 | 0.009 | 1.00 | 0.043 |
| Dusk–Night | −0.002 | 0.044 | 759 | −0.045 | 1.00 | 0.047 |
| Early–Morning | −0.107 | 0.035 | 766 | −3.091 | 0.025 | 0.017 |
| Early–Night | −0.109 | 0.048 | 756 | −2.289 | 0.199 | 0.027 |
| Morning–Night | −0.002 | 0.039 | 762 | −0.057 | 1.00 | 0.05 |
| Month | | | | | | |
| April–February | −0.119 | 0.029 | 76.2 | −4.002 | 0.003 | 0.002 * |
| April–January | −1.054 | 0.069 | 65.5 | −1.528 | 0.727 | 0.020 |
| April–July | −0.052 | 0.043 | 122.3 | −1.218 | 0.886 | 0.031 |
| April–June | 0.012 | 0.035 | 81.9 | 0.357 | 0.999 | 0.041 |
| April–March | −0.078 | 0.029 | 64.3 | −2.717 | 0.111 | 0.009 |
| April–May | −0.063 | 0.042 | 54.8 | −1.514 | 0.735 | 0.021 |
| February–January | 0.014 | 0.068 | 68.9 | 0.209 | 1.00 | 0.048 |
| February–July | 0.068 | 0.042 | 161.1 | 1.604 | 0.680 | 0.017 |
| February–June | 0.132 | 0.035 | 121.5 | 3.810 | 0.004 | 0.005 * |
| February–March | 0.042 | 0.029 | 105.6 | 1.460 | 0.768 | 0.024 |
| February–May | 0.057 | 0.042 | 68.1 | 1.358 | 0.822 | 0.029 |
| January–July | 0.054 | 0.075 | 83.0 | 0.710 | 0.992 | 0.033 |
| January–June | 0.118 | 0.071 | 73.8 | 1.649 | 0.651 | 0.014 |
| January–March | 0.027 | 0.069 | 70.3 | 0.398 | 0.999 | 0.043 |
| January–May | 0.042 | 0.075 | 65.9 | 0.564 | 0.998 | 0.038 |
| July–June | 0.064 | 0.046 | 151.7 | 1.400 | 0.801 | 0.026 |
| July–March | −0.026 | 0.041 | 147.6 | −0.632 | 0.996 | 0.036 |
| July–May | −0.011 | 0.051 | 94.5 | −0.217 | 1.00 | 0.050 |
| June–March | −0.090 | 0.034 | 105.5 | −2.702 | 0.108 | 0.007 |
| June–May | −0.075 | 0.045 | 72.0 | −1.668 | 0.639 | 0.012 |
| March–May | 0.015 | 0.041 | 61.0 | 0.369 | 0.999 | 0.045 |

### 3.2. Analysis of Time—Activity Patterns

Figure 3 showed that when birds has outdoor access overnight, flamingos used enclosure zones that they were rarely seen in during the day (e.g., front grassed area). The highest percentage of courting birds in the early morning showed that the front grass was used for courtship display, then the islands in the pool and then into the water at dusk. Occurrences of foraging are high for all times of the day, but especially in the water during hours of darkness (Early and Night). The full output from the Poisson regression is found in the Supplementary Materials (Table S1) and this analysis showed that for each behaviour, there was a significant relationship between enclosure zone, time of day and month of observation on flamingo activity across daylight and nocturnal hours (Table 4).

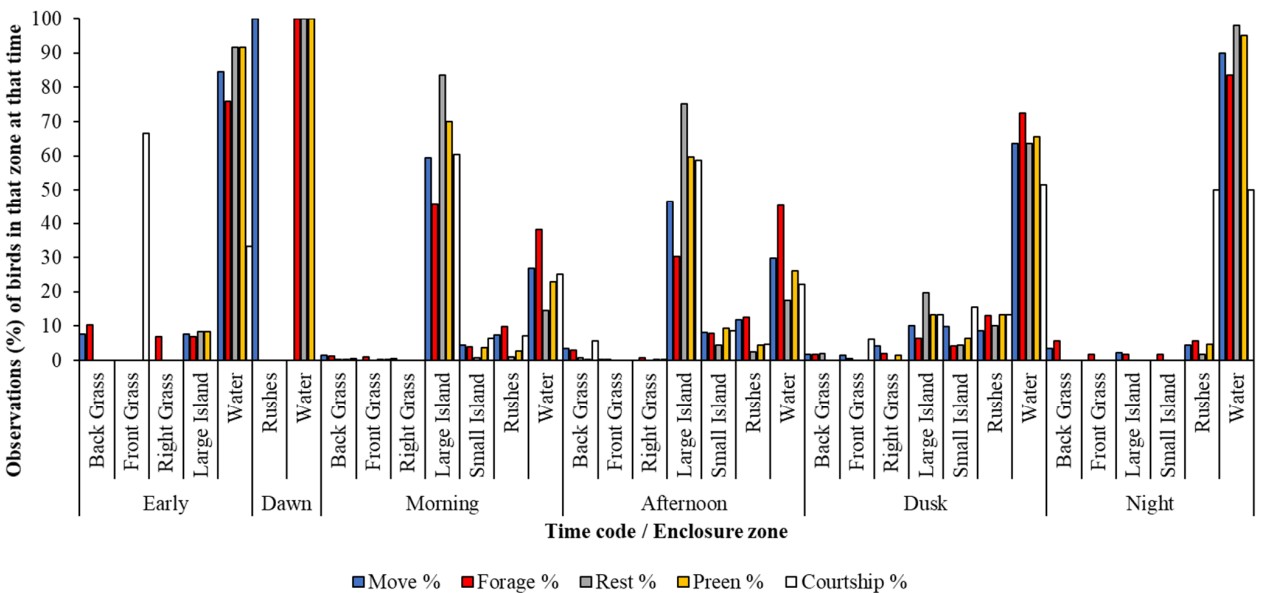

**Figure 3.** The percentage of outside observations (out of all observations) of flamingos performing five state behaviours within each enclosure zone at different times of the day.

**Table 4.** Wald's Chi-squared tests from Poisson regression on lesser flamingo behaviour for different times of day, different enclosure zones and different months.

| Behaviour | Predictor | Df | Chi-Squared Statistic | *p*-Value |
|---|---|---|---|---|
| Move | Zone | 6 | 876.89 | <0.001 |
| | Month | 4 | 306.83 | <0.001 |
| | Time | 5 | 610.74 | <0.001 |
| Forage | Zone | 6 | 1538.73 | <0.001 |
| | Month | 4 | 531.17 | <0.001 |
| | Time | 5 | 816.47 | <0.001 |
| Rest | Zone | 6 | 2163.67 | <0.001 |
| | Month | 4 | 540.89 | <0.001 |
| | Time | 5 | 1195.98 | <0.001 |
| Preen | Zone | 6 | 1014.24 | <0.001 |
| | Month | 4 | 351.02 | <0.001 |
| | Time | 5 | 625.63 | <0.001 |
| Courtship | Zone | 6 | 981.5 | <0.001 |
| | Month | 4 | 206.73 | <0.001 |
| | Time | 5 | 600.76 | <0.001 |

### 3.3. Indoor and Outdoor Enclosure Usage

Although there is little variation in SPI for use of inside or outside zones (Figure 4), flamingos used more enclosure zones at dusk when outside and had the most uneven enclosure usage at Dusk and at Night when inside. The range in the count of flamingos observed in each zone for each time of the day are provided Figures 5 and 6, and these figures identify differences in the total number of birds seen in each zone at each time. Flamingos indoors showed the widest range of number birds using the feeding, water, and indoor land areas in the Morning (Figure 5), whereas outdoors there are wide ranging numbers of birds using the water at Dawn and the large island in the Morning. For nocturnal observations (in the period Night), the largest range in number of birds was seen in the water (Figure 6).

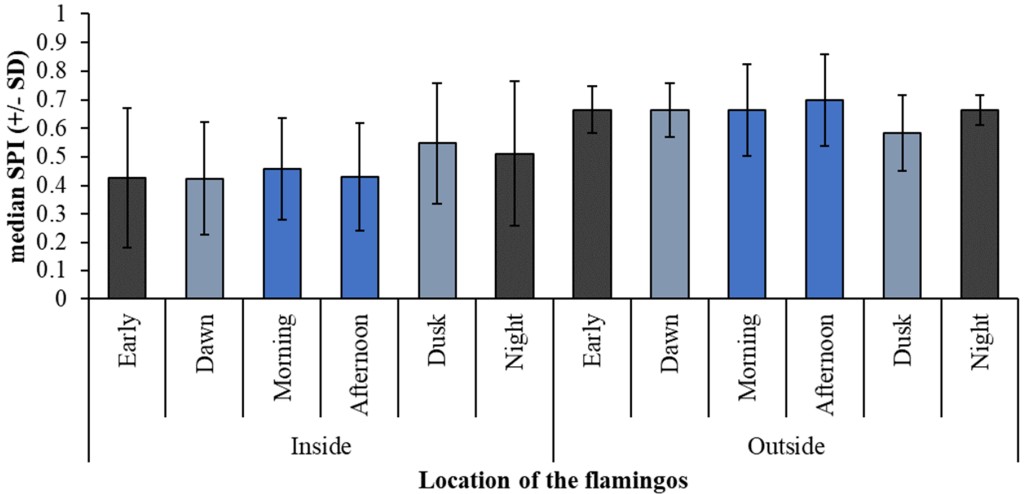

**Figure 4.** Zone occupancy at different times of the day (based on modified SPI) for when birds were observed inside or outside.

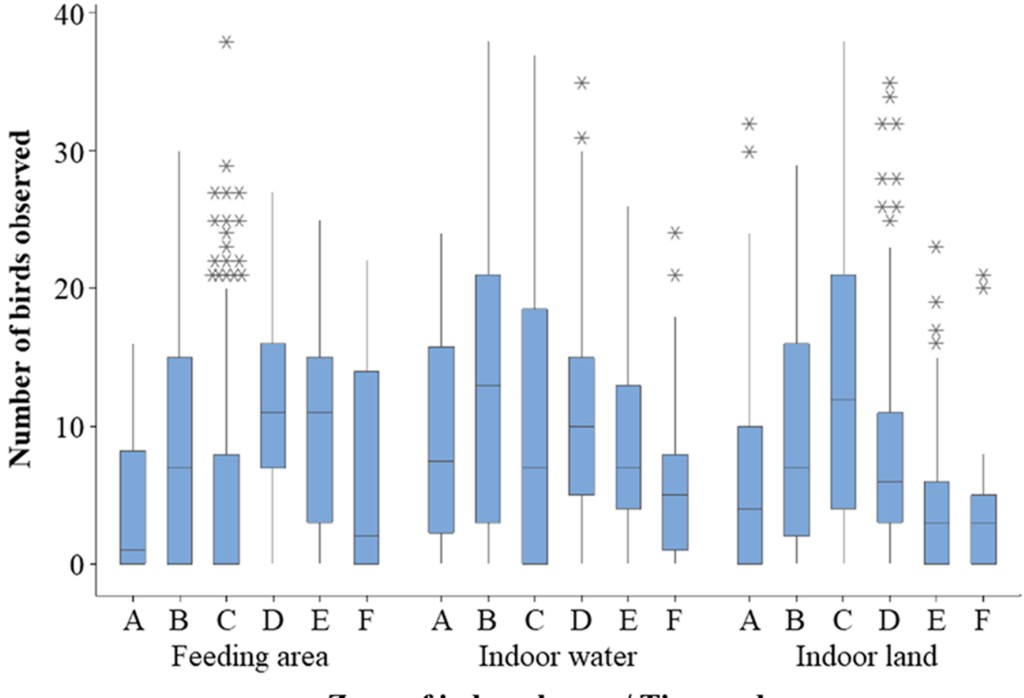

**Figure 5.** Count of birds in different indoor zones by time of day. A = Early; B = Dawn; C = Morning; D = Afternoon; E = Dusk; F = Night. * show outliers.

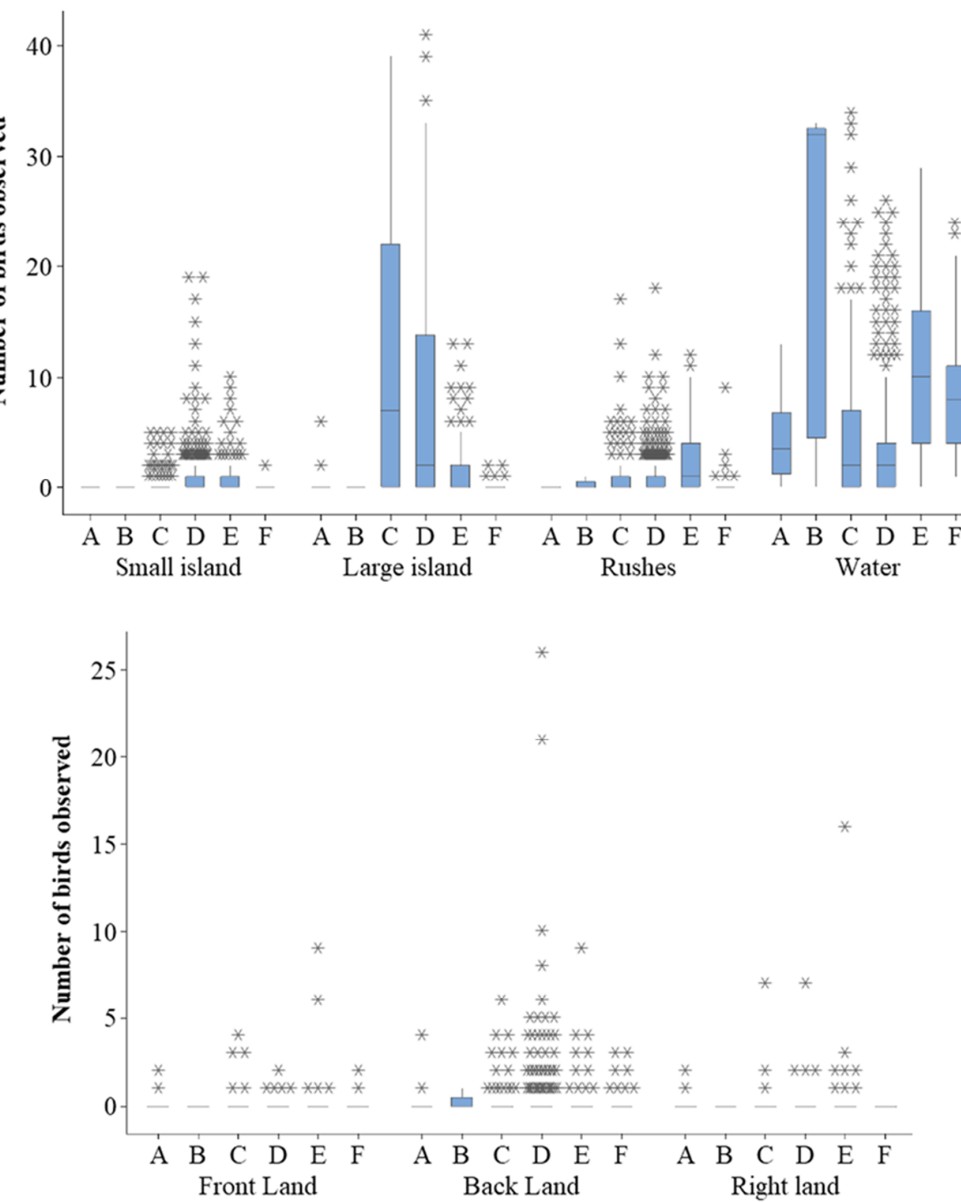

**Figure 6.** Count of birds in different outdoor enclosure zones by time of day. A = Early; B = Dawn; C = Morning; D = Afternoon; E = Dusk; F = Night. * show outliers.

For outdoors SPI time of day showed a significant relationship with enclosure usage ($F_{5, 728.2}$ = 15.121; $r^2$ = 21%; $p$ < 0.001) but month only showed a general trend ($F_{4, 56.7}$ = 2.453; $r^2$ = 21%; $p$ = 0.056). For indoor SPI, there was a significant difference in indoor SPI for month ($F_{6, 89.2}$ = 3.234; $r^2$ = 17%; $p$ = 0.006) and time of day ($F_{5, 862.1}$ = 3.149; $r^2$ = 17%; $p$ = 0.008). For outdoor SPI, Night and Dusk showed a significantly wider enclosure usage (lower SPI) when compared to Morning and Afternoon (higher SPI value), Figure 4. For indoor SPI, this pattern is reversed with Morning and Afternoon enclosure usage being wider (lower SPI) and Dusk enclosure usage being narrower (higher SPI), Figure 4. This may be reflected in Figure 5 that shows a wide range of birds at Dusk (point E on the graph) noted in the feeding area.

### 3.4. Indoor Enclosure Usage with and without a Keeper Present

The enclosure usage of these flamingos did not significantly differ when a keeper was present or absent (one sample sign test; n = 806; test median = 0.4853; $p$ = 0.149). However,

flamingo activity was significantly lower with no keeper present compared to when a keeper was in the house (one sample sign test; n = 778; test median = 1.0; $p < 0.001$).

## 4. Discussion

This research has shown that captive lesser flamingos can maintain high levels of activity during diurnal, crepuscular and nocturnal time periods. Higher activity at dusk, compared to other times of the day, was noted and this is worthy of further investigation. Wild flamingos may forage more actively during the evening and into the night in response to food availability [5] or when disturbances are likely to be more limited [16]. For example, nocturnal foraging may be a behavioural strategy to recover lost foraging time due to human disturbance [35]. Further research into any visitor effect on the foraging behaviour of captive flamingos and how this is impacted upon by differing degrees of visitor presence would illuminate any differences in diurnal versus nocturnal feeding rates. Movement patterns of wild lesser flamingos are noted as being nocturnal [36] and this need to move to new feeding patches or to nesting grounds may still exist in captive birds, maybe particularly so if these zoo-housed birds have wild origins. Patterns of inactivity in this captive flock of flamingos may reflect husbandry procedures, with inactivity seen at dawn preceding morning feeding of the birds. Birds were more likely to be inactive indoors during early periods if they had been active before midnight, and therefore were now roosting.

There was no defined pattern to enclosure usage for these lesser flamingos, with SPI values showing variation when birds were indoors and when they were in their outdoor enclosure. Flamingos used a range of enclosure areas at different times of the day and used these enclosure zones for specific behaviours. Outdoor islands and the outdoor pool were well utilised and were the site of a range of behaviour patterns displayed by these flamingos. Lesser flamingos are highly specialised birds [19], with specific foraging behaviours for the selection of microscopic aquatic plant material [13]. As such, the variation in enclosure usage and particularly the decisions that birds were making over the use of water bodies may reflect desires to perform foraging activity. Wild lesser flamingos display 10 specific individual feeding behaviours where the birds are either wading, swimming or up-ending to collect food [13]. Of these foraging behaviours, wild lesser flamingos change which are predominantly used during daytime and night-time [13] and further study of the type of nocturnal movement and foraging that captive flamingos perform (or are able to perform) should be implemented to further refine enclosures to meet the birds' ecological needs.

Large variation in the enclosure usage illustrated by Figures 4 and 5 may be a factor of the bird's social activities. A majority of flamingos choosing to perform the same behaviour as the other birds around them may have caused the outliers on these graphs at specific times. Captive flamingos will also move away from their main social group and act independently to conspecifics at different times of the day [37]. Consequently, the complex nature of flamingo social behaviour and the attraction strength of the main flock may be reflected in the variation expressed in this graph. Enclosure zoning (i.e., the number of zones defined and their size) can also influence calculated SPI values and therefore has a bearing on the accuracy of enclosure usage assessment [38]. Further research should consider the number of zones that a flamingo enclosure is divided into and compare different zone number within the same enclosure to evaluate the degree of variation in calculated SPI values.

The greater flamingos observed overnight at WWT Slimbridge Wetland Centre by Rose et al. [6] displayed more precise changes in activity and enclosure usage over time when compared to this flock of lesser flamingos. Differences in the ecology of the two species of bird, and the size of the flock and enclosure (larger in the case of the greater flamingos), and the age structure of the birds (a more uniform age of the lesser flamingos) may account for any differences. However, a key similarity of these two pieces of research was that they both evidence high rates of nocturnal activity in captive flamingos, and the use of water (the flock's pool) is particularly apparent during hours of darkness. Therefore, flamingos of

different species display nocturnal behavioural rhythms which should be catered for in the zoo.

Although month had no impact on flamingo activity in the outdoor enclosure, further measurement of weather and local climate (e.g., temperature) would be useful to understand any environmental influences over nocturnal enclosure usage. Captive flamingos are responsive to prevailing climatic conditions and can restrict activity and enclosure based on degree of sunshine, prevailing temperature or humidity [37]. Flamingos may remain indoors, or in sheltered areas of their exhibit during periods of inclement weather and this is likely to impact on their favoured enclosure areas as well as their partitioning of time-activity patterns. The siting of a flamingo enclosure within the zoo could be considered from a climate perspective, especially if encouraging courtship and nesting behaviour is a key goal of their presence in living collection, however more data are required to fully ascertain this.

This research has shown that captive lesser flamingos are disturbed by the presence of a keeper in the indoor house when birds are locked in that indoor house. Given the frequency of notifiable disease outbreak and the consequential need for birds to be locked indoors this is an important husbandry issue. Flamingo houses should be designed to ensure that birds can move away from animal care staff and not feel threatened by them. Further study should aim to understand the context of increased activity when animal care staff are present; for example, to decipher if flamingos were moving around more in an escape attempt or if there was evidence of displacement activity that could be used as a coping mechanism due to the closer-than-normal human presence. Although the presence of a keeper did not significantly impact on zone occupancy within the indoor house, it maybe that birds remained in specific areas of the house but showed more variation in behaviour during the period of time when animal care staff were present. Published research showed that zoo bird behavioural responses to human presence can vary by zoo and by enclosure. For example, two papers on penguin (Sphenisciformes) behaviour and pool use showed an increased use of the pool with increasing visitor number [39] in one article whereas the other highlights an opposite relationship [40]. This identifies the challenges and complexities in understanding any possible impact of human presence on zoo bird behaviour and enclosure utilisation.

*Limitations and Research Extensions*

The challenges of using camera traps to collect longitudinal behavioural data for groups of animals in the zoo are evident in this research. Further study should aim to place more camera traps in more areas of the enclosures of group living species, where many individuals could be out of range of the camera at any one time. Recording wild lesser flamingo nocturnal activity is also noted as a challenge [13] and therefore consideration of the type of camera, the number of cameras used and their range and picture quality is needed to ensure valid and robust data collection. Refining the recording of individual behaviour would enable identification of which birds were inside the house and outside in the main enclosure and this could be undertaken using high-definition video recording to enable the capture of individual bird identifiers, for example leg rings. Although this study has provided some useful information on what captive lesser flamingo do and where they go during the day and overnight, birds being hidden within or behind the flock or due to the quality of the footage has reduced the number of records of behaviour and space use and therefore the confidence in overall findings.

The main challenge during the completion of this research project was the impact of the *Avian Influenza* outbreak on bird housing and husbandry. Flamingos were housed indoors with no outdoor access until March 2018 and as shown in our research, this did not restrict activity levels with flamingos remaining active across a full 24 h cycle. Extended indoor housing is becoming more of a reality for many species of zoo birds as high pathogenic outbreak of *Avian Influenza* have affected zoos around the world since the inception of this research project [41]. Even when external environmental stimuli are missing, these

flamingos remained active and therefore consideration of their behavioural needs when indoors for 24 h per day is needed regarding future development and implementation of species-specific husbandry. Waterbirds appear to be particular susceptible to *Avian Influenza* virus [42,43] and they can be some of the species first housed indoors when *Avian Influenza* measures are announced [44]. We encourage other researchers to measure the impact of *Avian Influenza* and/or extended periods of indoor housing on the behaviour and welfare of other species of zoo bird as this is currently a gap in the evidence-basis for suitable zoo husbandry.

Despite these shortcomings, this study shows the relevance of remote technology to zoo animal behaviour and welfare study. Further development of techniques to collect data on species' (and individual) responses to captivity when animal care staff are not around are needed to gain baseline information on welfare states [45]. Refining data collection techniques using high quality camera traps or implementation of closed-circuit television (CCTV) systems, e.g., Brady et al. [46], to measure the behaviour of social or gregarious species across a 24 h period would improve our knowledge of animal needs and wants across their full circadian rhythm. Use of video recording and applicable software for analysis of video footage would also increase the accuracy of categorising different behaviours at each recording interval [47], and would therefore increase the amount of behavioural data available for analysis. It is important to remember that animals do not cease behaving because they have been taken off exhibit for the night and when animal care staff have left the zoo. Therefore, indoor housing or night quarters need to be developed with evidence on how animals behave overnight and/or when they are alone in the zoo to maintain good welfare across the full 24 h period.

Differences in the behavioural budget of species housed under human care when compared to wild data, specifically pertaining to nocturnal and diurnal activity are apparent. For example, captive golden hamsters (*Mesocricetus auratus*) are more likely to be nocturnal in their activity patterns whereas wild animals are generally active during the morning and later afternoon [48]. This evident temporal behavioural difference in a species that we are very familiar with should galvanise further research into the influence of time of day on the myriad of species that we house with which we are less familiar. Whilst numerous environmental factors may override internal time-keeping mechanisms, such as predation, local climate or food availability [49], comparison of time-activity patterns for zoo-housed species across different facilities and evaluation against data from free-living populations would provide understanding on the biological relevance of nocturnal activity patterns. Given the use of remote camera traps to measure the responses of wild animals to their environment, and to the pressures they may face from humans [50], coordinated activities of zoo-based and field scientists could facilitate the collection of 24 h behavioural data to help evaluate captive animal circadian rhythms.

Environmental enrichment can promote beneficial, biologically relevant behaviour patterns and improve welfare of captive flamingos [51]. The animal's enclosure itself can be a source of enrichment [52,53] and in the case of these lesser flamingos this may be apparent. The larger island, with a substrate of sand and mud was used for foraging by these lesser flamingos predominantly during Morning and Afternoon. Wild lesser flamingos in East Africa will spend a large proportion of their time (dependent on the Rift Valley Lake they are occupying) skimming wet mud for microscopic algae and plant material [13]. Therefore, use of the island for foraging in the daytime and the water for foraging during hours of darkness may reflect a wild-type time-activity budget of these lesser flamingos and show the ecological suitability of the enclosure for this flock of captive birds. Further review of enclosure usage assessment, considering the effect of the group on the decisions made by individuals of such social species of when to use different zone areas [54] can be helpful in determining overall ecological suitability of zoo environments.

## 5. Conclusions

This flock of captive lesser flamingos displayed nocturnal activity patterns as has been identified in wild populations of this species, in wild populations of other flamingo species, and in other captive populations of flamingo too. These results are important for the development of bird husbandry protocols and provide more evidence for the complexity of flamingo activity patterns that should be catered for when these species are managed under human care. The indoor housing of flamingos and the environment they are provided with when indoors needs to consider this variation in activity over a 24 h cycle to allow for natural behaviours to be performed. This is relevant to short-term housing (e.g., overnight) as well as for period of prolonged time spent indoors (e.g., for biosecurity need or periods of inclement weather). Further development of how remote monitoring is used for the collection of behavioural data, combining webcam or CCTV data collection approaches from wild birds or at other zoological institutions, would be useful to our further understanding of the behavioural rhythm of this species. The change in behaviour patterns over different times of the day, and the use of specific enclosure areas (e.g., the sanded island for foraging during daylight hours) suggest the biological usefulness of this style of enclosure for these birds and we encourage other zoos holding lesser flamingos to provide a pool of differing depth and large, flat islands of sand and mud to enable an enriched existence for their birds.

**Supplementary Materials:** The following supporting information can be downloaded at: https://www.mdpi.com/article/10.3390/jzbg3040046/s1, Table S1: Poisson regression on the total counts of birds seen in each enclosure zone for each time code for each month of the study.

**Author Contributions:** Conceptualization, P.E.R.; methodology, P.E.R.; validation, P.E.R.; formal analysis, P.E.R.; investigation, J.C.; resources, P.E.R.; data curation, J.C., P.E.R. and J.E.B.; writing—original draft preparation, P.E.R.; writing—review and editing, J.C., J.E.B., L.M.R.; supervision, P.E.R. and L.M.R.; project administration, J.C. and P.E.R. All authors have read and agreed to the published version of the manuscript.

**Funding:** This research received no external funding.

**Institutional Review Board Statement:** Ethical approval for the project was provided by the Ethics Committee of University Centre Sparsholt and after review by WWT living collections teams and veterinary department as part of WWT's Animal Welfare & Ethics Committee.

**Data Availability Statement:** Raw data can be downloaded from https://doi.org/10.6084/m9.figshare.21546825 (accessed on 12 November 2022).

**Acknowledgments:** Thanks to Mark Roberts and Phil Tovey for their assistance with camera location and permission to install cameras in the flamingo house. Grateful thanks to Ruth Cromie and Baz Hughes for enabling the research to proceed at WWT Slimbridge.

**Conflicts of Interest:** The authors declare no conflict of interest.

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
