# Peer review of "What’s Black and White and Pink All Over? Lesser Flamingo Nocturnal Behaviour Captured by Remote Cameras"

_2673-5636, doi:10.3390/jzbg3040046_

Round 1

Reviewer 1 Report

The manuscript has a study focus of ample interests for animal studies, be it on welfare, husbandry or simply behavior studies. The issue is rather important, and the authors apply robust methods to understand the behavioral differences.

However, a few areas could receive some attention from the authors in order to increase the value and clarity of their manuscript, which I list below.

Is it safe to assume that size differences between the indoor and outdoor spaces of the enclosure are statistically negligible to disregard a possible effect of available space between settings?

The enclosure and camera setup deserves more detail. what are the dimensions of the enclosure, of the different environments (grass patch, interior...). the distance between cameras, and to any important landmarks within the enclosure. this will help to better understand the spatiality of the activity and the capacity of the cameras to cover the area. Information on the cameras (model, resolution and picture quality) would also help validate the method as providing sufficiently clear and reliable pictures is rather important for the study (given that pictures are a limitation per se).

Was video recording considered at any point during the sampling planning? I believe that video recording would decrease the frequency of unknown behaviors detected, as well as facilitate identification of behaviors in general.

the portion of the methods that discusses the statistics is a tad messy, especially with several calls back and forth on R/RStudio and the like. My suggestion would be leaving the software to be cited at the end of the session (both packages and the R apps used), there's no need to point which package did which exact analysis.

I would also suggest depositing the data and code used for the analysis at a repository (such as Zenodo or Dryad), as a second layer of information. with this, you can focus on a less obstructive methods section and have a good reference to direct people that are more curious about your study. there are free ones such as Zenodo that will easily allow you to deposit the data and add value to your study.

Author Response

Reviewer 1

The manuscript has a study focus of ample interests for animal studies, be it on welfare, husbandry or simply behavior studies. The issue is rather important, and the authors apply robust methods to understand the behavioral differences. However, a few areas could receive some attention from the authors in order to increase the value and clarity of their manuscript, which I list below.

Thank you for the useful and helpful comments to further develop the paper. We have provided responses to and explained edits for each point raised.

Is it safe to assume that size differences between the indoor and outdoor spaces of the enclosure are statistically negligible to disregard a possible effect of available space between settings?

Thank you for the comment. We are struggling to understand the edit here? Data have been analysed independently for indoor and outdoor areas. We do not attempt to compare what the flamingos did inside, with what they did outside because of differences in space availability. We are presenting how the birds used their space and what they did in this when they were seen indoors and when they were viewed outdoors. If the reviewer could provide more information on their point here, we would happily attempt an edit of the relevant part of the manuscript.

The enclosure and camera setup deserves more detail. what are the dimensions of the enclosure, of the different environments (grass patch, interior...). the distance between cameras, and to any important landmarks within the enclosure. this will help to better understand the spatiality of the activity and the capacity of the cameras to cover the area. Information on the cameras (model, resolution and picture quality) would also help validate the method as providing sufficiently clear and reliable pictures is rather important for the study (given that pictures are a limitation per se).

Thank you for the comment. We have already provided the overall size of the enclosure (m2) and we have given the size of each enclosure zone. We have provided further details on the range of each camera (as a distance to a fixed point within the enclosure) and we have provided the maximum width and length of the outside enclosure and the length of the indoor house.

We have also already provided the camera model in the text and the quality of photo. However, we have expanded the information on the cameras used by including extra information from the manufacturer.

Was video recording considered at any point during the sampling planning? I believe that video recording would decrease the frequency of unknown behaviors detected, as well as facilitate identification of behaviors in general.

Thank you for the suggested improvement, we have included the use of video recording as a potential project extension in the discussion. Due to the location of the enclosure and the house, and the challenge of connecting cameras to mains electricity for long-term recording, we used still photos from these trail cameras as they were battery powered.

the portion of the methods that discusses the statistics is a tad messy, especially with several calls back and forth on R/RStudio and the like. My suggestion would be leaving the software to be cited at the end of the session (both packages and the R apps used), there's no need to point which package did which exact analysis.

Thank you for the comment. We have re-worded the data analysis section of the methods slightly to accommodate this suggestion, however, when the paper was originally submitted the editor recommended the inclusion of the specific detail pertaining to each test, how it was run, in what software and why. Therefore, we feel it pertinent to include these details attached to each piece of inferential analysis.

I would also suggest depositing the data and code used for the analysis at a repository (such as Zenodo or Dryad), as a second layer of information. with this, you can focus on a less obstructive methods section and have a good reference to direct people that are more curious about your study. there are free ones such as Zenodo that will easily allow you to deposit the data and add value to your study.

Thank you for the comment, we have uploaded the raw data files to figshare and have provided a DOI https://doi.org/10.6084/m9.figshare.21546825. We have quoted the link in the text.

Reviewer 2 Report

The authors have described the behavior of captive lesser flamingos. This is an important addition to scarce similar research. The methodology has been described in detail and then analysed adequately. Findings have been extensively discussed and their usefulness for managing captive flamingos has been highlighted. The manuscript is very well written and relevant for the journal.

Comments

Line 50 – Rose et al. [15]

Line 72 – Tindle et al. [7]

Line 82 – Rose et al. [6]

Lines 124-126, 189, 265-268, 287-288, 319-320, 325-326, 357-358, 362, 386-387, 390-391 - Figure and Table captions should not be italicized. Please revie and follow journal style.

Line 191 and after – Line spacing is not the same as before. Please amend.

Line 226 – subset…

Line 448 – Rose et al. [6]

Line 584 – an enriched…

Author Response

Reviewer 2

The authors have described the behavior of captive lesser flamingos. This is an important addition to scarce similar research. The methodology has been described in detail and then analysed adequately. Findings have been extensively discussed and their usefulness for managing captive flamingos has been highlighted. The manuscript is very well written and relevant for the journal.

Thank you for the feedback. We are pleased that you feel the manuscript is a useful addition to the journal and worthy of publication.

Comments

Line 50 – Rose et al. [15]

We have edited to remove the comma.

Line 72 – Tindle et al. [7]

Edited to remove extra authors in the text.

Line 82 – Rose et al. [6]

Edited accordingly

Lines 124-126, 189, 265-268, 287-288, 319-320, 325-326, 357-358, 362, 386-387, 390-391 - Figure and Table captions should not be italicized. Please revie and follow journal style.

Thank you for the comment. We have reviewed all table and figure captions and amended accordingly.

Line 191 and after – Line spacing is not the same as before. Please amend.

Apologies for the change in line spacing format. This has now been corrected.

Line 226 – subset…

Edited

Line 448 – Rose et al. [6]

Edited accordingly

Line 584 – an enriched…

Edited